# Enhanced Adsorption, Photocatalytic Degradation Efficiency of Phenol Red Using CuZnAl Hydrotalcite Synthesized by Co-Precipitation Technique

**Van Nhuong Vu [1,\*], Thi Ha Thanh Pham [1], Quoc Dung Nguyen [1], Thi Hau Vu [1], Thi Tu Anh Duong [1], Thi Hue Tran [1] and Thi Kim Ngan Tran [2,\*]**

[1] Faculty of Chemistry, Thai Nguyen University of Education, No. 20 Luong Ngoc Quyen Street, Thai Nguyen City 24000, Vietnam
[2] Institute of Applied Technology and Sustainable Development, Nguyen Tat Thanh University, Ho Chi Minh City 70000, Vietnam
\* Correspondence: nhuongvv@tnue.edu.vn (V.N.V.); nganttk@ntt.edu.vn (T.K.N.T.); Tel.: +84-765-712-086 (T.K.N.T.)

**Abstract:** $ZnAlCO_3$ hydrotalcite materials modified by $Cu^{2+}$ ions were synthesized by the co-precipitation method according to the molar ratios of $(Cu^{2+} + Zn^{2+}):Al^{3+}$ as 7:3. Thus, the modified materials contain various molar ratios of $Cu^{2+}$ from 0–3.5 in the samples. The synthesized materials were characterized by X-ray diffraction pattern (XRD), FT–IR, EDS, SEM, the $N_2$ adsorption/desorption isotherm (BET), and UV–Vis DRS spectrum. The synthesized materials were characterized by a layered double hydroxide structure—such as hydrotalcite. The specific surface area BET increases slightly, corresponding to the increasing $Cu^{2+}$ molar ratios, and the bandgap energy Eg decreases accordingly. Especially, these material samples have a high phenol red (PR) adsorption capacity at a concentration of 100 ppm and PR was degraded under a 30 W LED light with over 90% of conversion efficiency in the presence of 1.2 mL of 30% $H_2O_2$ solution. In addition, the CuH–3.5 material sample maintained stability after four times catalytic reuse. Therefore, this material can be used as an effective treatment for the wastewater of the sedge mat weaving village.

**Keywords:** co-precipitation; photocatalyst; degradation; phenol red; characterization

## 1. Introduction

In recent years, layered double hydroxide materials (LDHs) and hydrotalcites have been of great scientific interest as they are able to memorize structures (memory effect) [1–3], are easily synthesized by the co-precipitation method, and exhibit adsorption and catalytic properties, hence a wide range of applications such as wastewater treatment, catalysis, photocatalysis, pharmaceuticals, organic synthesis and antibacterial agents [4–6]. Hydrotalcite materials have the formula of $Mg_{0.667}Al_{0.333}(OH)_2(CO_3)_{0.167}(H_2O)_{0.5}$ in the natural minerals. Many different layered double hydroxides and hydrotalcite materials have successfully synthesized for their applications [1,7–11]. Besides $MgAlCO_3$ hydrotalcites, $ZnAlCO_3$ materials—such as hydrotalcites—are also interesting to scientists [4,5,11–14]. Many authors have synthesized the $ZnAlCO_3$ materials modified with various factors such as Zn:Al molar ratio, metal salt precursors, medium pH, gel aging temperature, interlayered anion type, and the molar ratio of modified metal ions. Most of the hydrotalcite materials are synthesized by the co-precipitation method. The results showed that these synthesized materials are capable of effectively decomposing sustainable organic compounds and dyes in water. However, some limitations remained, including (the first) the poor adsorption capacity of these materials for dyes, especially cationic dyes, (the second) the high processing costs due to the use of expensive and power-consuming equipment such as UV [3,9], Xenon [5,10,15], and halogen [16] lamps as irradiation lights, and (the third) only survey on the low concentration of dyes.

The $ZnAlCO_3$ hydrotalcite materials modified by $Cu^{2+}$ ions proved the best photocatalytic activity with the molar ratios of $Cu^{2+}$ ion in the range of 2.5–3.5 [17]. These materials had a high photocatalytic activity in the decomposition reaction of rhodamine-B and pigments in textile dyeing wastewater. However, the adsorption and photocatalytic degradation capacity of these materials for anionic dyes (for example, phenol red), the efficiency of catalyst reuse, the factors improving the activity of the materials, and the dyes mineralization ability of these materials have not been studied. Therefore, we have synthesized $ZnAlCO_3$ material samples modified by different molar ratios of $Cu^{2+}$ (0, 2.0, 2.5, 3.0, and 3.5) by the co-precipitation method. The synthesized materials were characterized and investigated for the adsorption and decomposition capacity for PR and treatment of pigments in sedge textile dyeing wastewater under mild conditions with ambient temperature, medium pH of weak acidic range, and a 30 W LED light (Philips, Amsterdam, Netherlands, $\lambda > 400$ nm, $\lambda_{max} = 464$ nm). In addition, we investigated various factors that affect the PR degradation in the water of modified $ZnAlCO_3$ hydrotalcite materials, including concentrations of PR and $H_2O_2$, medium pH, and material reuse performance.

This paper is the first to present the results of the evaluation of PR adsorption and degradation ability at high concentrations (50, 75, and 100 ppm) using $ZnAlCO_3$ hydrotalcites modified by $Cu^{2+}$ ions and a 30 W LED lamp as the visible light source. In addition, the research results also identified important parameters enhancing the catalytic activity of the synthesized material samples.

## 2. Materials and Methods

### 2.1. Preparation of Chemicals

The chemicals were used to prepare materials all having a purity of more than 99%. Zincat nitrate $(Zn(NO_3)_2 \cdot 6H_2O)$ and aluminum nitrate $(Al(NO_3)_3 \cdot 9H_2O)$ were purchased from Xilong Scientific, Shanghai, China. Copper nitrate $(Cu(NO_3)_2 \cdot 3H_2O)$, sodium carbonate $(Na_2CO_3)$, sodium hydroxide (NaOH), phenol red (PR), and 30% hydrogen peroxide solution $(H_2O_2)$ were purchased from Merck, Darmstadt, Germany.

### 2.2. Material Synthesis

Five samples of hydrotalcites modified by $Cu^{2+}$ ion (denoted as CuH–n) with the molar ratios of $Cu^{2+}$ ion, respectively, 0, 2.0, 2.5, 3.0, and 3.5 were synthesized by the co-precipitation method. The process of synthesizing materials is summarized in Figure 1 and performed as described in the article [17].

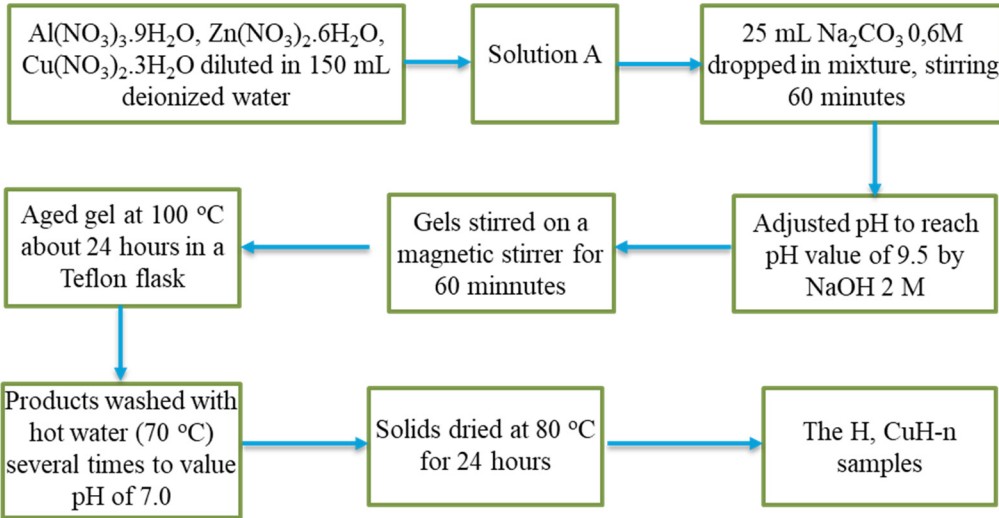

**Figure 1.** The synthesis stages of 5 material samples H, CuH–2.0, CuH–2.5, CuH–3.0, and CuH–3.5.

### 2.3. Analysis of Structure and Catalytic Properties of Materials

The crystalline phase composition of the synthesized samples was determined by the XRD pattern on the D8—ADVANCE 5005—Brucker instrument (Bruker AXS, Karlsruhe, Germany). UV–Vis DRS spectra were determined on a HITACHI U-4100 instrument (Tokyo, Japan). The specific surface area was determined by the $N_2$ adsorption–desorption isotherm method (BET) on a MicroActive for TriStar II Plus Version 2.03 instrument (Micromeritics, Norcross, GA, USA). SEM images of the representative material samples were measured using a Hitachi S–4800 instrument (Tokyo, Japan). The FT–IR spectra of the two samples H, CuH–3.5 were characterized using L1600400 Spectrum Two DTGS (Perkin Elmer Co., Waltham, MA, USA).

### 2.4. Adsorption and Photocatalytic Degradation Ability of Synthetic Materials for Phenol Red and Dyes in Wastewater

The adsorption capacity in the dark of the synthesized materials for PR was investigated using 0.2 g of samples mixed with 250 mL of solution PR at different concentrations (50, 75, and 100 ppm). The reaction mixtures were wrapped in black-colored bags and stirred on a magnetic stirrer at 400 rpm for 120 min at room temperature (28 ± 2 °C) to reach adsorption equilibrium. After every 30 min interval, a volume of PR solution was removed from the reactor (5–7 mL of PR solution), centrifuged, and diluted until the concentration of PR was within the limits of the calibration curve. The PR adsorption efficiency according to adsorption time was determined by Equation (1):

$$H\%_{ads} = \frac{C_0 - C_t}{C_0} \times 100 (\%)$$

(1)

where $C_0$ is the initial concentration of PR and $C_t$ is the concentration of PR at the time of the survey. In addition, the time to reach the adsorption equilibrium of the material can be determined under the survey conditions. The survey experiments were conducted 3 times. The results were calculated to obtain average values and used to evaluate the PR adsorption activity of the investigated materials.

After the mixture had reached the adsorption equilibrium, 1.2 mL of 30% $H_2O_2$ was added to the reaction mixture to evaluate the ability to decompose PR under the 30 W LED light. After every 30 min interval, 5–7 mL of the reaction mixture was removed, centrifuged, and the absorbance of PR was measured at 435 nm. The RP degradation efficiency was determined using Formula (2):

$$H\%_{degrad} = \frac{C_0 - C_t}{C_0} \times 100 (\%)$$

(2)

where $C_0$ is the initial concentration of PR and $C_t$ is the concentration of PR at the survey times. The survey experiments were conducted 3 times. The results were calculated to obtain average values and used to evaluate the photocatalytic activity of the investigated materials.

Based on the obtained results, the sample with high PR adsorption capacity was selected to investigate the effects of several important factors, including the concentrations of PR (50, 75, and 100 ppm) and $H_2O_2$ (0, 0.05, 0.1, 0.15, 0.20, 0.25, and 0.42 M), and the medium pH values. The medium pH of the 100 ppm PR solution was adjusted by HCl 0.1 M and NaOH 0.1 M solutions from the initial pH value (4.25) to 2.0, 6.0, 8.0, 10.0, and 12.0. After stirring for 120 min in the dark to reach the adsorption equilibration at investigated pH value, after every period of 30 min of illumination, the mixture was centrifuged to remove the catalyst and adjusted to the initial pH value (approximately 4.25) and diluted ten times to determine the concentration of PR remaining in the reaction mixture. In addition, we have studied the reuse performance of photocatalyst and applied material to treat sedge mat textile dyeing wastewater. The survey experiments were conducted

3 times. The results were calculated to obtain average values and used to evaluate the PR photocatalytic activity of the investigated materials.

### 2.5. Determination of PR Concentration in Water

The concentration of PR in water was determined by the standard curve method. The absorbance of PR was measured at 435 nm by a Shimadzu UV1700 instrument (Tokyo, Japan). The standard curve equation for determining the concentration of PR in water is $y = 0.0592x + 0.0015$ ($R^2 = 0.9998$) in Figure 2.

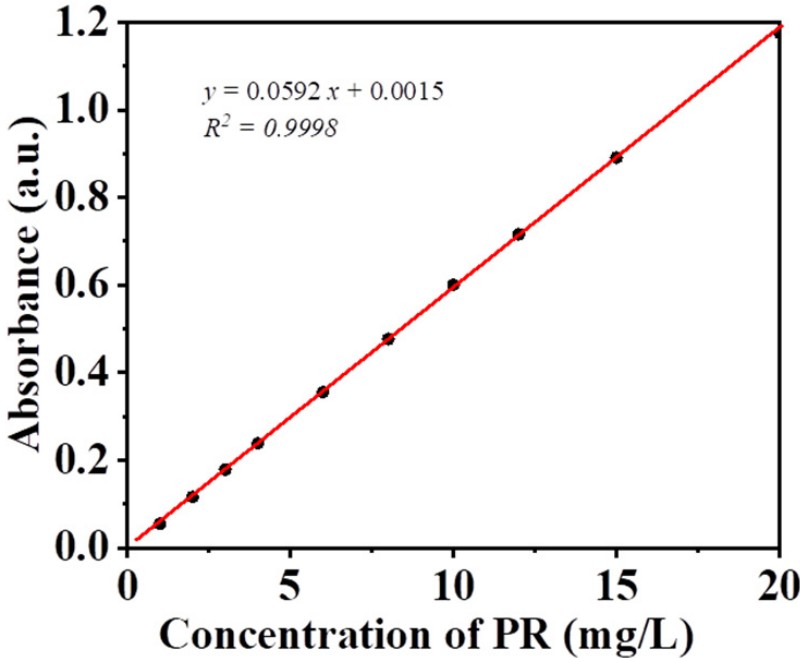

**Figure 2.** Standard curve for determination of PR concentration in water.

### 2.6. Application of Material for Textile Dyeing Wastewater Treatment

The ability of the synthesized CuH–3.5 sample to decompose pigments in the composition of sedge mat textile dyeing wastewater was investigated in a sedge mat weaving village located in Dong Bang village, An Le commune, Quynh Phu district, Thai Binh Province, Vietnam. The initial wastewater sample was diluted 15 times with distilled water, then the pH value was adjusted from 4.01 (initial medium pH) to 6.0 and 8.0 with HCl 0.1 M and NaOH 0.1 M solutions. Next step, 0.2 g of CuH–3.5 sample and 5.0 mL of 30% $H_2O_2$ were added into three beakers containing 250 mL of adjusted pH wastewater sample. After every 60 min of irradiation, the mixtures were sampled, centrifugated, diluted ten times by distillation water, and scanned spectra on a UV–Vis 1700 instrument. The survey experiments were conducted three times.

The adsorption and decomposition efficiency of pigments in the sedge mats textile dyeing wastewater composition were calculated according to the following two Formulas (3) and (4):

$$H\%_{ads} = \frac{Abs_0 - Abs_t}{Abs_0} \times 100 \, (\%) \tag{3}$$

$$H\%_{degrad} = \frac{Abs_0 - Abs_t}{Abs_0} \times 100 \, (\%) \tag{4}$$

where $Abs_0$ is the absorbance of the pigments at 545 nm at the initial time and $Abs_t$ is the absorbance of the pigments at 545 nm at the time of the investigation.

### 2.7. Determination of Chemical Oxygen Demand (COD)

After the illumination periods to decompose the pigments in the textile dyeing wastewater, the wastewater was centrifuged, and then used to determine the COD index according to the lighting time.

The COD value was determined by adding 2.5 mL of wastewater sample into the COD digestion cuvette. In the first step, the decomposition of excess $H_2O_2$ was carried out by adjusting the solution pH with NaOH 20% to a value pH of 10.0, then refluxing at 100 °C for 30 min on a COD digester (Hanna HI839800–02). Next step, 1.5 mL $K_2Cr_2O_7$ 0,4 N and 3.5 mL of mixture of concentrated $H_2SO_4$ and $Ag_2SO_4$ solutions were added into the COD digestion cuvette. After that, the cuvette was placed in the COD digester for 2 h reflux at 150 °C. In the final step, the absorbance of the mixture was measured by a Shimadzu UV1700 instrument at 605 nm. The percentage results of dyes mineralization in wastewater were calculated using the following Equation (5).

$$\%\mathrm{COD} = \frac{\mathrm{COD}_0 - \mathrm{COD}_t}{\mathrm{COD}_0} \times 100\,(\%) \tag{5}$$

where $\mathrm{COD}_0$ is the COD value at the initial time without lighting and $\mathrm{COD}_t$ is the COD value at each survey time.

The COD index was determined by the dichromate method. The absorbance was measured by a Shimadzu UV1700 instrument at 605 nm. The standard curve equation for determining the COD index is $y = 0.0003x - 0.0008$ ($R^2 = 0.9991$) (Figure 3).

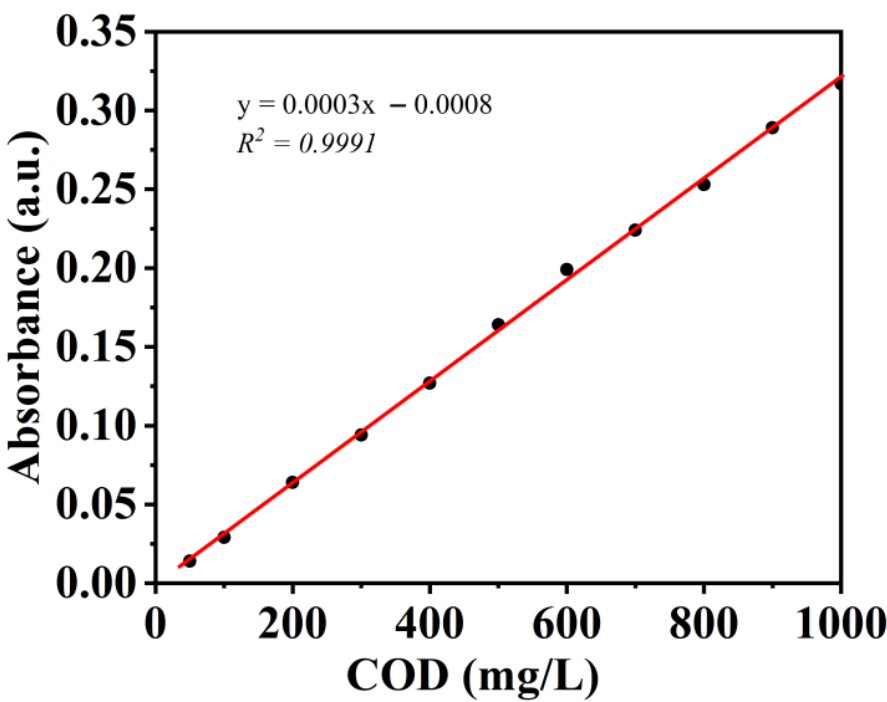

**Figure 3.** Standard curve for determination of COD index.

### 3. Results and Discussion

#### 3.1. Characterization of Synthesized Materials

3.1.1. X–ray Diffraction Patterns (XRD) and UV–Vis DRS Spectra of the Synthesized Samples

XRD patterns and UV–Vis DRS spectra of the hydrotalcites have been presented in a previous study [17]. Therefore, in this paper, we summarized some basic characteristics of the synthesized materials in the XRD patterns and UV–Vis DRS spectra. The XRD patterns confirmed that the synthesized samples had a hydrotalcite-like structure. However, there was a decrease in the hydrotalcite structure when $Cu^{2+}$ ions were implanted into the brucite network. The basic spacing ($d_{003}$) of the samples ranged from 7.73–8.56 Å. No crystalline

phases of CuO appeared even though the four CuH–n samples were all black in color. The parameter "a" ranged from 3.058 to 3.074 Å, and parameter "c" reached 23.01 to 24.44 Å. The average grain sizes calculated using the Scherrer formula decreased from 33.28 to 15.09 nm.

It can be seen that changes in lattice parameters will lead to some consequences as follows: Lattice parameters a and c characterize the distance between metal cations between lattice layers and the thickness of the brucite layer, and the distance between the inner layers. When parameters a and c are changed, the layered structure order of hydrotalcite is degraded, leading to a change in structural characteristics and changes in physico-chemical properties. For example, the change of lattice parameters a and c led to the formation of hydrotalcite sheets of $Cu^{2+}$ ion-modified materials that were more uniform and more ordered than $ZnAlCO_3$ hydrotalcite, and a marked shift of the absorption edge into the visible region, and at the same time, reducing the band gap energy of the modified materials. In addition, this change also causes the mesoporous system to appear in the modified material samples. These changes are shown through the results of structural characterization of the materials, as well as clearly shown through the adsorption capacity as well as the photochemical degradation of phenol red and colorants in the wastewater of the synthetic materials in the below survey results.

The modification of hydrotalcite with $Cu^{2+}$ ions yielded new materials in which $Cu^{2+}$ ions have been homomorphically replaced with $Zn^{2+}$ ions in the brucite lattice. This is shown through the results of the XRD histogram analysis and UV–Vis DRS spectrum. Parameters a and c only change in a narrow range, so the synthesized materials still maintain a hydrotalcite-like layer structure. In addition, the UV–Vis DRS spectra of the CuH–n samples did not appear the peak absorption peak of CuO in the large wavelength region (>700 nm). Thus, it can be seen that there is the isomorphic substitution of $Cu^{2+}$ ions in the brucite lattice.

The UV–Vis DRS spectra of the materials showed that their absorption edges shifted strongly to the visible light region when increasing the ratio of $Cu^{2+}$ in the material samples from 0 to 3.5. The bandgap energy Eg of the H and CuH–n samples decreased from 3.02 eV (H) to 2.3 eV (CuH–3.5). This result indicates that the samples modified by $Cu^{2+}$ possess good photocatalytic activity under visible light. These things proved that $Cu^{2+}$ ions have the role of shifting the absorption edge of modified hydrotalcite material samples to the visible region, and at the same time, reducing the band gap energy of the material. This is explained by the charge transfer from $O^{2-}$ ions to $Cu^{2+}$ ions in the tetrahedral medium and the charge displacement between the d–d orbitals in the octahedral environment of $Cu^{2+}$ ions.

### 3.1.2. FT–IR Spectra of Material Samples

The FT–IR spectra of the two representative samples H and CuH–3.5 (Figure 4) showed a broad peak found at about 3448 $cm^{-1}$ due to the bond stretching vibration modes of the hydroxyl groups (OH), both in the layers and the $H_2O$ molecules in the interlayer. The band at 1609.30 $cm^{-1}$ was due to the deformation vibration of $H_2O$ molecules in the interlayer. The bands at 1377.86 and 1361.23 $cm^{-1}$ were the stretching vibrations of interlayered carbonate ions. The lower wavenumber bands (from 431.84 to 828.89 $cm^{-1}$) were due to vibrations of the metal–oxygen lattice involving the layer cations ($Cu^{2+}$, $Zn^{2+}$, and $Al^{3+}$): M–O, M–OH, O–M–O, and M–O–M. In particular, the sample CuH–3.5 also appeared with vibration at 828.89 cm $^{-1}$ which was attached to $Cu^{2+}$ containing LDH. These results revealed that the product is a carbonate-type hydrotalcite compound and the $Cu^{2+}$ ions were present in the brucite network of the materials [1,6,8,9,13,18,19].

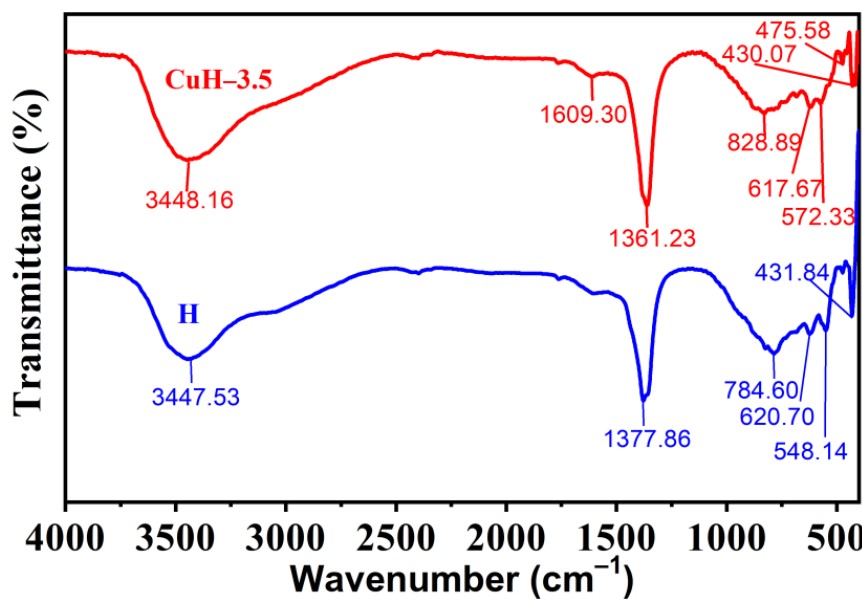

**Figure 4.** FT–IR spectra of the H and CuH–3.5.

### 3.1.3. EDX Spectra and SEM Images of the H and CuH–3.5

The results of the EDX spectra of the two representative material samples H, CuH–3.5 in Figure 5a,b confirmed the presence of three elements O, Zn, and Al in the H sample and four elements O, Al, Cu, and Zn in the CuH–3.5 sample. The atomic percent compositions of O, Al, and Zn elements in the H sample are 72.19, 8.94, and 18.87%, respectively, corresponding to the molar ratio of Zn:Al as 18.87:8.94 fitted with 2.11:1 (Figure 5a). The CuH–3.5 sample has an atomic percent composition of O, Al, Cu, and Zn elements of 75.76, 8.88, 7.82, and 7.53, respectively (Figure 5b). In this modified material sample, the percentage ratio of Cu:Zn atoms is 7.82:7.53, appropriately with 1.04:1, and the ratio of Zn:Al is 7.53:8.88, appropriately with 1:1.18. These results proved that the molar ratio of Cu:Zn was quite suitable as compared to the amount of $Cu^{2+}$ and $Zn^{2+}$ selected to synthesize the CuH–3.5 sample. However, the molar ratio of Zn:Al in the CuH–3.5 material sample decreased sharply from 2.11:1 in the H sample to 1:1.18 in the CuH–3.5 sample, leading to a decrease in the hydrotalcite structure when increasing the amount of $Cu^{2+}$ in the bruxit matrix. This result was consistent with the XRD patterns of the hydrotalcites above.

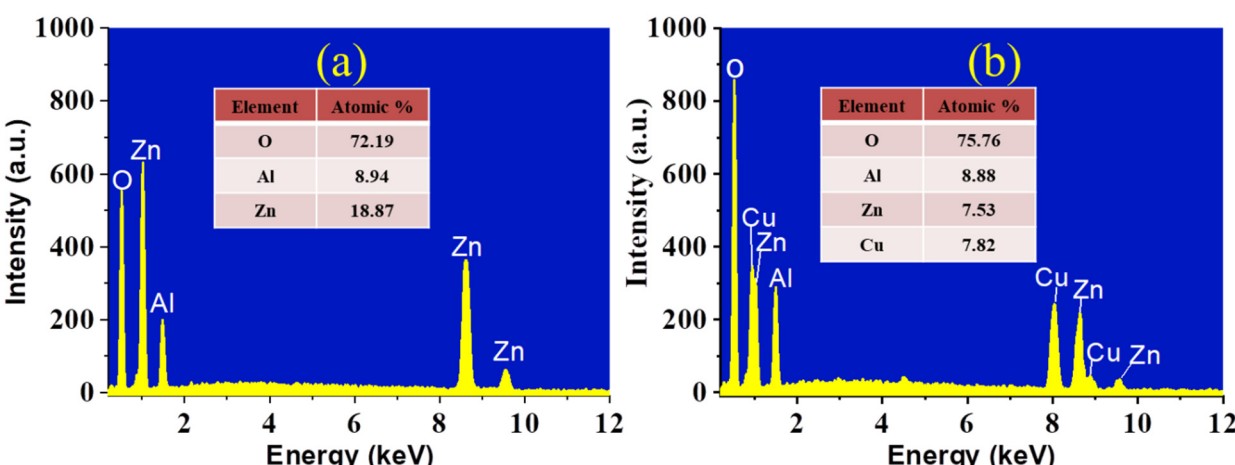

**Figure 5.** *Cont.*

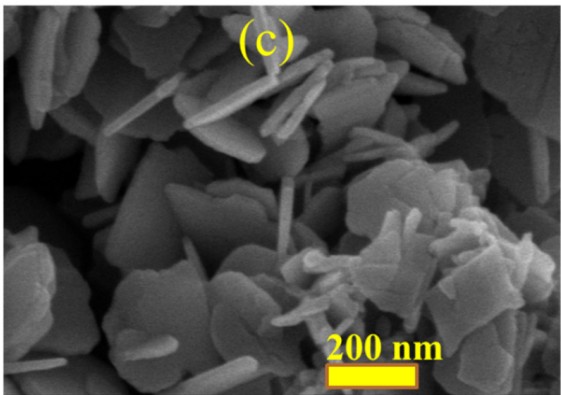
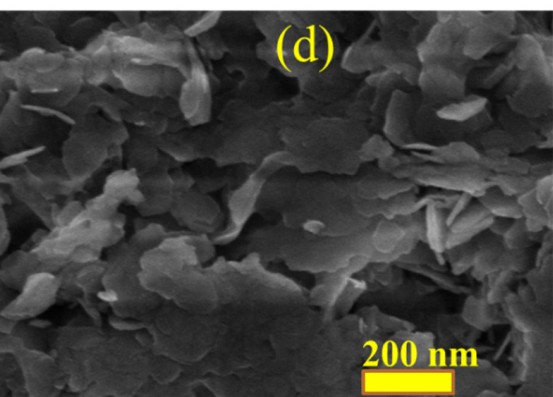

**Figure 5.** EDX spectra of the H (**a**), CuH–3.5 (**b**) and SEM images of the H (**c**), CuH–3.5 (**d**).

SEM images of two material samples of H and CuH–3.5 also showed a change in the size of the hydrotalcite sheets (Figure 5c,d). The H material sample had larger slabs, while the hydrotalcite sheets of the CuH–3.5 material were more uniform and smaller in size. Therefore, the CuH–3.5 material sample was predicted to have a more developed surface and a larger surface area than the H.

### 3.1.4. $N_2$ Adsorption/Desorption Isotherms (BET) of Hydrotalcite Samples

The $N_2$ adsorption and desorption isotherms of five hydrotalcite samples (Figure 6) show that the hydrotalcites have $N_2$ adsorption and desorption curves of type IV and H3 type, respectively, according to IUPAC classification [1,6,10,20,21]. The porous diameters and porous volumes of the five samples range from 15.8 to 37.9 nm and 0.091 to 0.208 cm$^3$/g, respectively, which were typical for medium porous materials (Table 1). The porous diameter and porous volume change non-linearly with an increasing molar ratio of Cu$^{2+}$ ions in the samples from 0 to 3.5. On the contrary, Brunauer–Emmett–Teller (BET) specific surface area increased uniformly and linearly from 16.1 to 37.1 m$^2$/g. As a result of the increased surface area (BET), the modified CuH–n materials become more porous than the H sample. This result was consistent with the results of the SEM image of the two representative samples above. Normally, the BET-specific surface area decreased with the increasing molar ratio of the modified metal ion. Therefore, the BET surface area was the smallest value when the molar ratio of the modified metal ions was the highest value [1,8,21,22]. This could be explained by the decrease in the layered double structure of the hydrotalcite and the aggregation and growth of the crystals with the increase in the amount of modified metal ions in the sample. However, in certain cases, the BET surface area increased with the increasing molar ratio of modified metal ions [10]. If we carefully observe the adsorption and desorption isotherms $N_2$ of the 5 composites in Figure 6, we can see a change in the shape, as well as an expanded hysteresis loop of the adsorption and desorption $N_2$ curves. All three material samples CuH–2.5, CuH–3.0, CuH–3.5 have isotherms converting from H3 type to H4 type of type IV according to IUPAC classification. Therefore, these three samples have a medium capillary pore structure, which leads to an increase in the specific surface area BET. The increasing surface area is an important factor when predicting the adsorption capacity of synthesized samples.

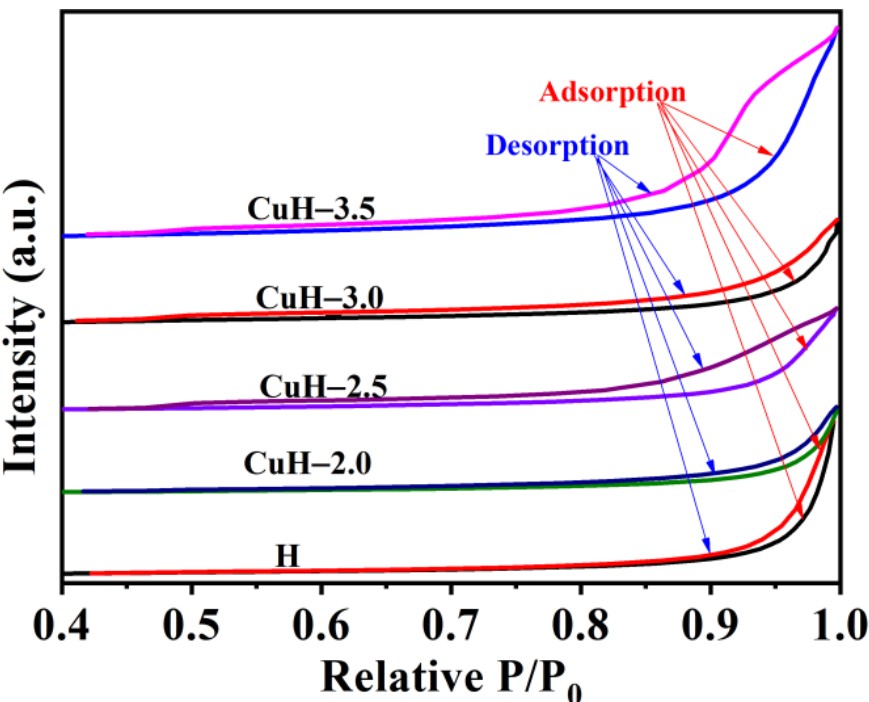

**Figure 6.** $N_2$ adsorption/desorption isotherms of five hydrotalcite samples.

**Table 1.** Specific surface area (BET), capillary diameter, and volume capillaries of five composites.

| No. | Surface Area BET ($m^2$/g) | Porous Diameter (nm) | Porous Volume ($cm^3$/g) |
|---|---|---|---|
| H | 16.1 | 37.6 | 0.151 |
| CuH–2.0 | 17.5 | 22.9 | 0.100 |
| CuH–2.5 | 26.7 | 15.8 | 0.091 |
| CuH–3.0 | 33.9 | 24.5 | 0.208 |
| CuH–3.5 | 37.1 | 19.2 | 0.178 |

*3.2. PR Adsorption, Degradation Ability on Synthesized Material Samples*

3.2.1. Survey Results on the PR Adsorption Ability in the Dark

The adsorption capacity of five hydrotalcites in the dark was investigated for PR concentrations of 50, 75, and 100 ppm. As shown in Figure 7, it can be seen that the hydrotalcites had a good adsorption capacity for PR in the order of CuH–3.0 > CuH–3.5 > CuH–2.5 > CuH–2.0 >> H. The PR adsorption efficiency can reach about 70% on the CuH–3.0 sample at a PR concentration of 50 ppm. When increasing the concentration of PR from 50 to 100 ppm, the PR adsorption efficiency of the investigated materials decreased sharply. For instance, the PR adsorption efficiency decreased from 70.7–48.1–37.2% after 180 min of stirring in the dark for the CuH–3.0 material of 50–75–100 ppm, respectively. Furthermore, no significant change in the PR adsorption efficiency was observed after about 120 min of stirring in the dark. Therefore, 120 min was the adsorption equilibrium time of the materials.

PR is known as a pH color indicator in cell biology laboratories and as an organic dye [23,24]. PR has the colors purple and pink in basic medium and yellow in acidic medium. PR exists in three forms (i.e., $H_2^+PS^-$, $HPS^-$, and $PS^{2-}$), depending on medium pH [25]. In addition, PR also contains a negatively charged sulphate radical in any medium. Therefore, similarly to MO (methyl orange), PR also acts as an anionic dye. LDHs materials have a good anion adsorption capacity due to their ability to exchange anions between $CO_3^{2-}$ ions with MO anions or due to the electrostatic interaction of $SO_3^{2-}$ ions of colorants

with metal ions in the interlayered of brucite [2,10,26,27]. In addition, the structural memory effect of LDHs plays an important role in the adsorption process of the anion dyes [27]. Besides the hydrotalcite-like laminar structure, with $CO_3^{2-}$ ion in the interlayers, the H and CuH–n composites also have a mesoporous structure with the specific surface area increased from H to CuH–3.5 sample. So, it is possible to store PR molecules in these porous systems. The reason is that the molecular size of PR is about 15 Å (1.5 nm), so it can exchange ions with $CO_3^{2-}$ ions and be trapped in the capillary systems of the materials [25].

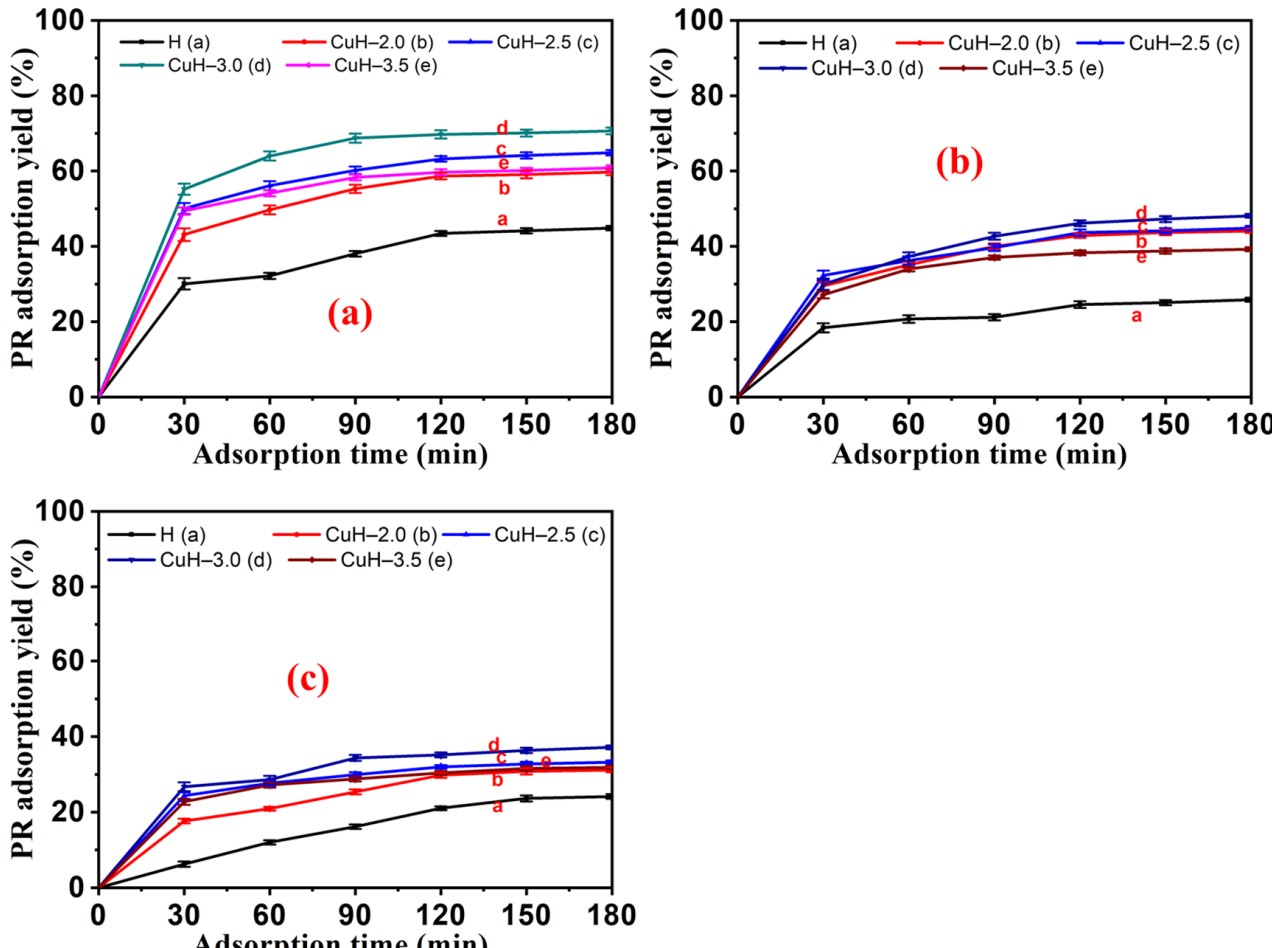

**Figure 7.** PR adsorption capacity at concentrations of 50 ppm (**a**), 75 ppm (**b**), and 100 ppm (**c**) on hydrotalcite samples.

### 3.2.2. The Ability to Degradation of PR under Visible Light

Effect of Irradiation Time and Ratio of $Cu^{2+}$ in Samples at Different Concentrations of PR

The photocatalytic activity of five synthesized materials in the decomposition reaction of PR was investigated under the visible light of a 30 W LED (λ > 400 nm, $\lambda_{max}$ = 464 nm). The efficiency of the RP adsorption and decomposition process was calculated according to Formulas (1) and (2), respectively.

Figure 8 shows that the H and CuH–n materials exhibited a high PR degradation ability. The efficiency of PR degradation depends on the irradiation time, PR concentration, and molar ratio of $Cu^{2+}$ in the samples. The degradation efficiency of PR decreased with increasing the PR concentration from 50 to 100 ppm. The yield of the PR conversion can reach 90–93% after only 60 min of illumination with a PR concentration of 50 ppm on all four CuH–n samples. However, the yield of the PR conversion reduced to only 85–87% on CuH–3.0 and CuH–3.5 samples after 240 min of illumination at a PR concentration of 100 ppm. The survey results showed that the photocatalytic activity was in decreasing order: CuH–

3.5 > CuH–3.0 > CuH–2.5 > CuH–2.0 >> H. These results were completely consistent with the UV–Vis spectrum DRS and activation energy Eg of the five synthesized samples.

**Figure 8.** PR adsorption and photocatalysis process at concentrations of 50 ppm (**a**), 75 ppm (**b**), and 100 ppm (**c**) on the hydrotalcites.

The photocatalytic activity of H and CuH–n materials can be explained by the decrease in the Eg bandgap energy with increasing the $Cu^{2+}$ amount in the sample and the catalytic role of $Cu^{2+}$ ions in photogenerating $e^-$–$h^+$ pairs, thus generating hydroxyl radicals ($^{\bullet}OH$) in the presence of $H_2O_2$ [1,10,21,28]. The photocatalysis processes to degrade PR are proved by Equations (6)–(10) below:

$$Cu^{2+}\text{-ZnAl} + h\nu \rightarrow Cu^{2+}\text{-ZnAl} (e^-, h^+) \tag{6}$$

$$h^+ + H_2O_2 \rightarrow 2^{\bullet}OH \tag{7}$$

$$H_2O_2 + e^- \rightarrow {}^{\bullet}OH + OH^- \tag{8}$$

$$OH^{\bullet} + RP \rightarrow \text{colorless reduction products} \tag{9}$$

$$h^+ + RP \rightarrow \text{colorless oxidation products.} \tag{10}$$

The UV–Vis spectra showed that the absorption peak of PR at 435 nm decreased after 120 min of stirring in the dark (Figure 9). Under irradiation, the absorption peak of PR at 435 nm decreased sharply for the first 30 min, then slowly disappeared after 210 min, as indicated by the reaction mixture gradually becoming colorless. These results showed that the conjugate systems of the chromogenic groups were completely converted into colorless intermediates. In addition, the absorption peak at about 265 nm characterized by a benzene

nucleus was also degraded within only 30 min of illumination. This result confirmed the effective color treatment ability of CuH–n materials. Especially, the synthesized materials are capable of degrading PR at a high concentration of 100 ppm with an efficiency of over 90% after 240 min of illumination with a 30 W LED lamp.

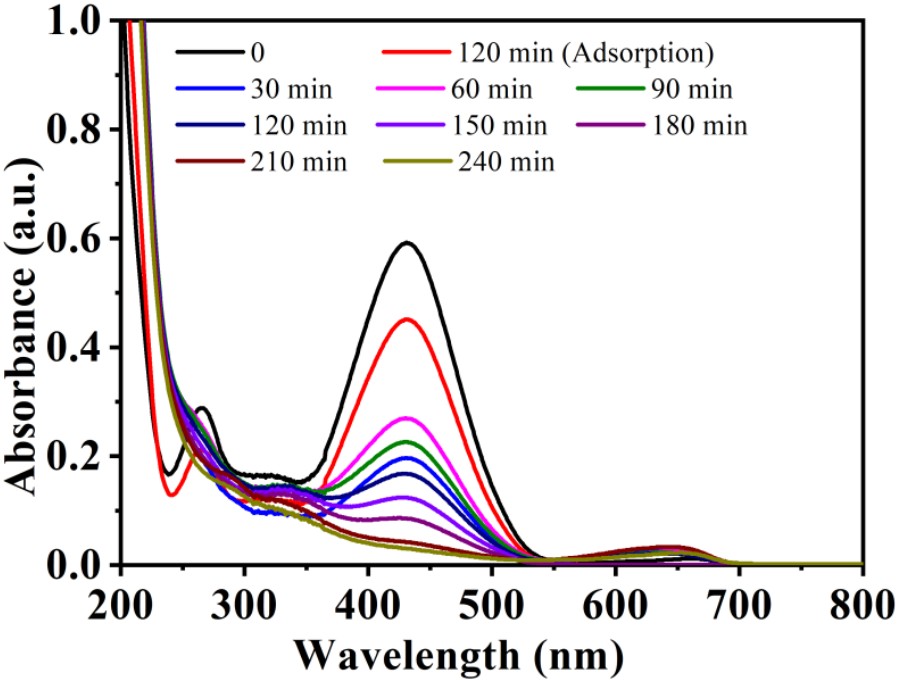

**Figure 9.** UV–Vis spectra of PR after 240 min of irradiation process by 30 W LED on the CuH–3.5 sample. Note: 250 mL of the 100 ppm PR solution, 0.2 g CuH–3.5 sample, 1.2 mL of 30% $H_2O_2$, solution diluted 10 times.

Effect of $H_2O_2$ Concentration

The CuH–3.5 sample and PR at a concentration of 100 ppm were selected to further investigate other factors that affect the catalytic activity of the material, including the concentration of $H_2O_2$, and medium pH value.

It can be seen from Figure 10 that the concentration of $H_2O_2$ (0–0.42 M) significantly affected the photocatalytic activity of the investigated material sample. In the absence of $H_2O_2$, the PR decomposition efficiency slightly changed by approximately 10%, and the adsorption activity occurred mainly with an efficiency of 35.2%. In the presence of $H_2O_2$ and the CuH–3.5 material, the PR decomposition efficiency increased rapidly as the $H_2O_2$ concentration increased from 0.05 to 0.42 M. The PR decomposition yield could noticeably reach about 90% after only 60 min of irradiation in the concentration of 0.42 M $H_2O_2$. However, as the $H_2O_2$ concentration continued to increase excessively beyond 0.42 M, the PR decomposition efficiency tended to decrease gradually. The reason was that as the $H_2O_2$ concentration becomes too large, the $OH^-$ radicals reacted again with $H_2O_2$ to produce $HOO^\bullet$ radicals that were less active than $OH^\bullet$ radicals, reducing the photocatalytic activity of the material [29,30]. In addition, the removal of excessive $H_2O_2$ was difficult.

Overall, the results have shown that the combination of the $H_2O_2$ agent with the material and 30 W LED light source improved the photocatalytic activity of the material, thereby enabling PR degradation at a high concentration of 100 ppm within a short period of time.

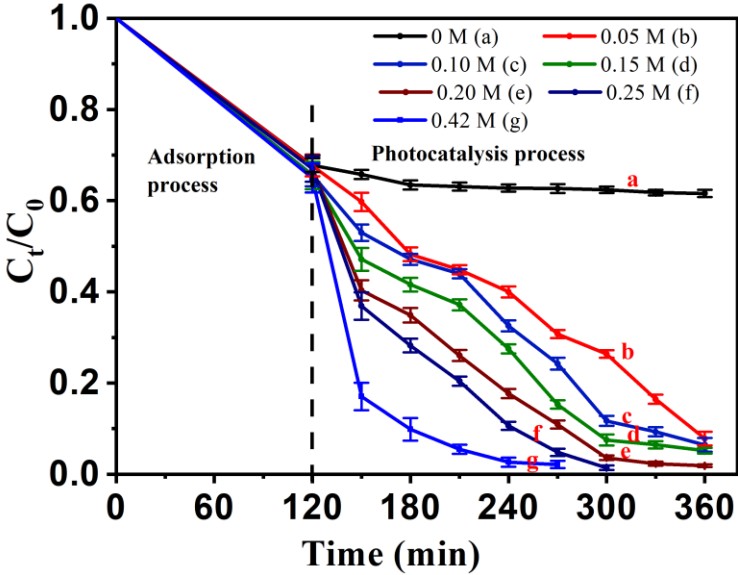

**Figure 10.** The ability to degrade RP 100 ppm of CuH–3.5 at different $H_2O_2$ concentrations.

Effect of Medium pH

The CuH–3.5 sample and $H_2O_2$ 0.05 M were used to investigate the influence of different pH values on the photocatalytic activity of the material. Figure 11 shows that the medium pH influenced clearly the photocatalytic activity of the CuH–3.5 material sample. The highest photocatalytic activity of the CuH–3.5 sample was achieved at the initial pH value of 100 ppm PR solution ($pH_{initial}$ = 4.25), while decreasing at other tested medium pH values, especially at pH = 2.0 and pH = 10.0–12.0. Such a reduction in the photocatalytic activity of the material can be explained due to the structural degradation of the material at low pH values (pH < 4.0) and the increasing solution viscosity at high pH values (pH = 10.0–12.0) [26]. The adsorption capacity of the material also decreased due to the competition between the anions of PR and $OH^-$ groups. In addition, the number of $OH^{\bullet}$ radicals was reduced due to the reaction of $OH^-$ ions with $H_2O_2$. All these causes reduced the photocatalytic activity of the material at the high medium pH. Therefore, the optimal pH range for the degradation of PR is selected as the weak acidic medium pH ranging from 4.0 to 6.0.

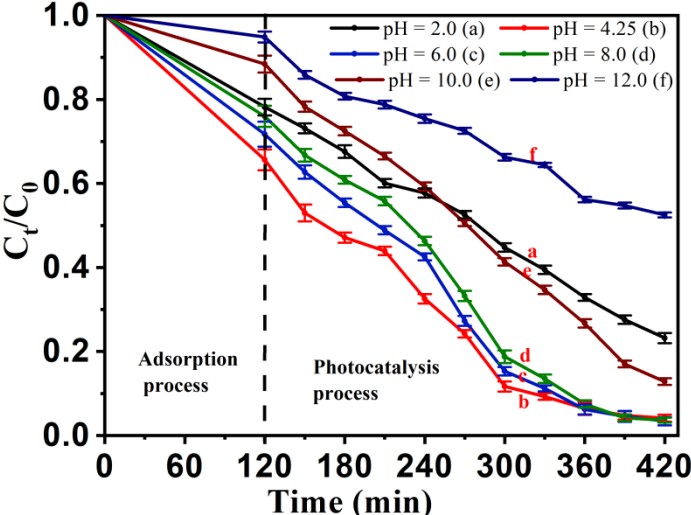

**Figure 11.** The decomposition of PR (100 ppm) by CuH–3.5 sample at different medium pH values.

Reusability of the Catalyst

The CuH–3.5 sample, PR concentration of 100 ppm, and $H_2O_2$ concentration of 0.1 M were used to investigate the reusability of the material. The efficiency of PR adsorption and decomposition after four times of catalyst reuse is shown in Figure 12.

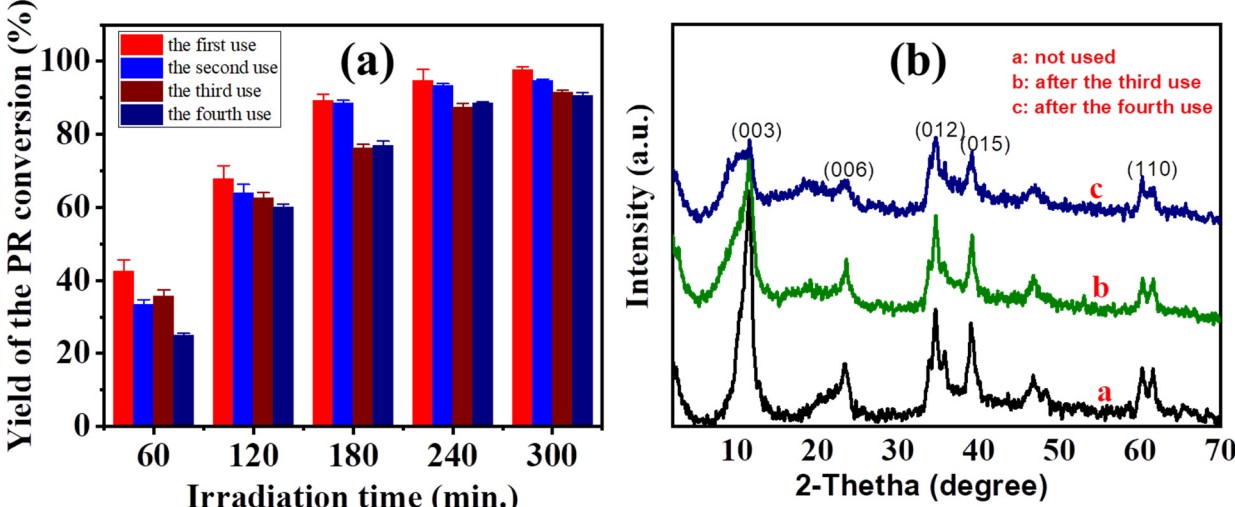

**Figure 12.** The efficiency of PR degradation after four times of catalyst reuse (**a**) and XRD patterns of CuH–3.5 material after four reuses (**b**).

After four times of reuse of the catalyst material, it can be seen that the PR decomposition capacity of the CuH–3.5 material sample reduced negligibly (Figure 12a). The conversion efficiency of 100 ppm PR after each reuse was about 90–97% after 300 min of illumination, which can be explained by the structural memory effect of the hydrotalcite materials. The material structure was slightly changed after three reused times of catalyst, and the stability of the material was well maintained (Figure 12b). These are similar to the study by Chen et al. [15] and Pan et al. [28]. Figure 12b shows the XRD patterns of the original CuH–3.5 sample after four reuse times. After three reuse times, the XRD pattern of CuH–3.5 showed no significant difference from the original CuH–3.5. However, there is a strong decrease in diffraction peak intensity $d_{003}$ characterized by a hydrotalcite lamellar structure after four reuse times. This is probably related to the partial leaching of $Cu^{2+}$ in solution, which degrades the hydrotalcite structure of the material [11].

### 3.3. Application of Materials for Textile Dyeing Wastewater Treatment

3.3.1. Investigation of the Ability to Decompose Pigments of the Synthesized CuH–3.5

The adsorption efficiency of dyes in textile dyeing wastewater on CuH–3.5 reduced from 26.7 to 15% in correspondence to the increasing pH after 120 min of adsorption in the dark to reach adsorption equilibrium Equation (3). The color degradation efficiency was measured at 545 nm, which gradually increased with the lighting time at all three investigated pH values in Equation (4). After 5 h of lighting, the degradation efficiency of dyes in wastewater can reach 97–98% at pH = 4.01 and 6.0 (Figure 13a). At the same time, the deep red color of the original pigment mixture turned into a lighter red color (Figure 13b). The results confirmed the applicability of CuH–n hydrotalcites to treat textile dyeing wastewater in practice.

The UV–Vis spectra of the pigments in the wastewater composition at pH = 6.0 changed over time (Figure 13c). The dyes in wastewater absorb in three wavelength regions: The first absorption region was at about 539–559 nm with an absorption maximum of about 545 nm, the second absorption was in the range of 370–500 nm, and the third absorption region at about 270 nm characterized by the benzene nucleus. During the illumination process, all three absorption peaks decreased sharply with the illumination time. The chromogenic systems in the region of 370–500 nm were broken more easily than in the 539–559 nm region.

After about 5 h of illumination, the absorption peaks in these absorption regions completely disappeared. The solution had a marked decrease in color intensity, as compared to the color of the original solution.

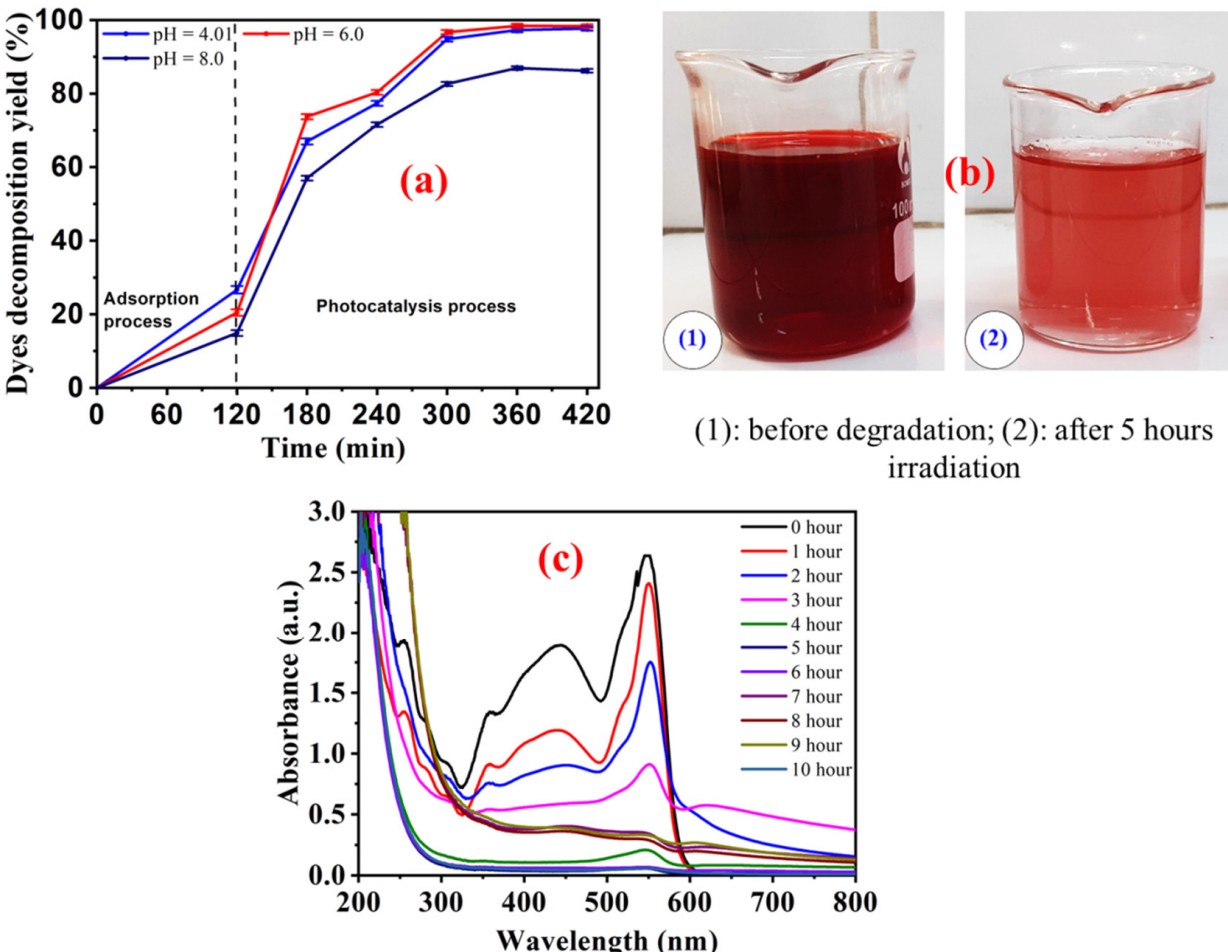

**Figure 13.** The efficiency of the dyes decomposition in sedge textile dyeing wastewater using CuH–3.5 sample (**a**), visual properties of the dyes before degradation (1) and after five hours of irradiation (2) (**b**), UV–Vis spectra of dyes after ten hours of lighting by the 30 W LED at medium pH value as 6.0 (**c**).

### 3.3.2. The Mineralization Capacity of the CuH–3.5 Material

As shown in Figure 14, the mineralization of wastewater reached about 88% after 12 h of treatment progress (includes 2 h of adsorption and 10 h of light). In particular, the rate of mineralization increased steadily over about 10 h of light. Because the composition of textile dyeing wastewater was quite complicated, the mineralization efficiency of organic matter in wastewater was only about 86–88% after 10 h of irradiation. In contrast, the conversion of pigments can reach 98% after only 5 h of lighting.

Overall, from the obtained results of textile dyeing wastewater treatment, the H materials modified by $Cu^{2+}$ have shown practical potential in treating textile dyeing wastewater at a low cost with the use of a 30 W LED light source, thereby reducing the cost of production energy equipment, as compared to other light sources (e.g., UV, xenon, and halogen).

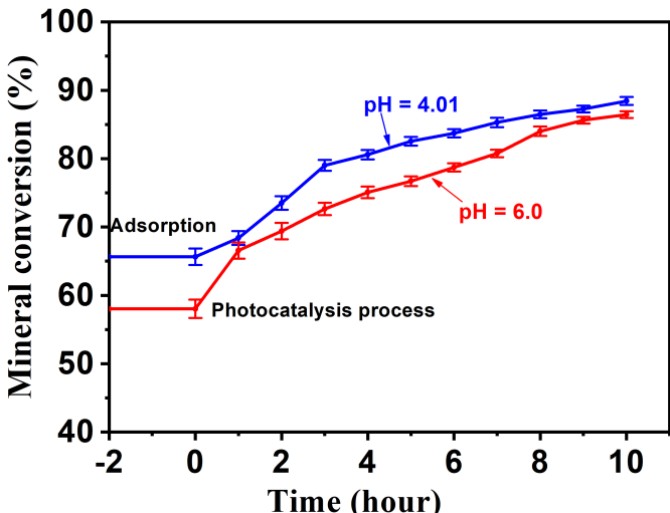

**Figure 14.** The mineralization conversion of the textile dyeing wastewater after 10 h of irradiation in the presence of CuH–3.5 material.

## 4. Conclusions

In the present study, H and CuH–n hydrotalcite materials were synthesized by the co-precipitation method. The characteristics of the layered structure were similar to hydrotalcite. However, the presence of $Cu^{2+}$ ions in the Bruxite network changed and reduced the lamellar structure of the hydrotalcite. Due to the lamellar structure being reduced, the CuH–n samples had a transformation from H3 type IV to H4 type according to IUPAC classification, which is characterized by mesoporous systems. The BET surface area increased when the molar ratio of $Cu^{2+}$ was raised from 0 to 3.5, increasing the adsorption capacity of PR. The adsorption of PR on the hydrotalcites could be attributed to the ion exchange mechanism between PR and carbonate ions in the interlayer and the medium capillary structure of the modified materials. The CuH–n hydrotalcites had absorption edges shifted strongly to the visible light region, causing the bandgap energy to decrease sharply from 3.02 to 2.3 eV. Due to a larger surface area and narrower bandgap Eg than H, both the CuH–3.0 and CuH–3.5 materials were capable of adsorbing and decomposing PR at high concentrations in the presence of $H_2O_2$. The PR adsorption efficiency can reach about 70% on the CuH–3.0 sample at a PR concentration of 50 ppm. The adsorption equilibrium of the materials for PR was achieved at about 120 min. The combination of $Cu^{2+}$ molar ratios of 3.0–3.5, medium pH range of 4.0–6.0, and a minor $H_2O_2$ dosage significantly enhanced the catalytic activity of the synthesized materials. In addition, the CuH–3.5 sample demonstrated structural stability, and this material's catalytic activity was maintained after four times of reuse. Furthermore, the high efficiency in the colorants treatment of sedge textile dyeing wastewater and the high mineralization of organic substances in the wastewater composition displayed the potential application of the synthesized CuH–3.5 in practice.

**Author Contributions:** Writing—original draft preparation, V.N.V. and T.H.T.P.; data curation, T.H.T.P. and Q.D.N.; formal analysis, T.H.T. and V.N.V.; conceptualization, T.H.V. and T.T.A.D.; methodology, Q.D.N. and T.K.N.T.; formal analysis, T.H.V. and T.T.A.D.; writing—review and editing, V.N.V. and T.K.N.T. All authors have read and agreed to the published version of the manuscript.

**Funding:** The authors would like to sincerely thank the financial resources from the project CS.2021.23 of the University of Education, Thai Nguyen University.

**Institutional Review Board Statement:** Not applicable.

**Informed Consent Statement:** Not applicable.

**Data Availability Statement:** All the data are available within the manuscript.

**Conflicts of Interest:** The authors declare no conflict of interest.

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
