# Peer review of "Enhanced Adsorption, Photocatalytic Degradation Efficiency of Phenol Red Using CuZnAl Hydrotalcite Synthesized by Co-Precipitation Technique"

_processes, doi:10.3390/pr10081555_

Round 1

Reviewer 1 Report

The manuscript "Enhanced adsorption, photocatalytic degradation efficiency of phenol red using CuZnAl hydrotalcite synthesized by co-precipitation technique" deals with the efficiency of hydrotalcite compound for adsorbing and photodegradation of phenol red in terms of Cu content. In general, the manuscript is well organized and it is interesting, Nonetheless, some key points need to be reviewed as follows.

1. Introduction section. The novelty and scientific contribution of the work must be stated clearly in this section.

2. Section 2.1 is missed. Please provide the chemicals that were used.

3. The sentence just before equation 3 must be eliminated.

4. Please amend the word adsortion in equation 3.

5. Section 3.1.1 XRD results. What would be the conclusion from the changes in the lattice parameters? Is there a substitutional insertion or an interstitial one?

6. Section 3.1.2 UV-vis results. Why does the Cu2+ reduce the Eg? Please provide a fundamental explanation besides the description of the result.

7. What the authors attribute the shift of the band of the interlayered carbonate to?

8. The Y axe in  Figures 8, 10 and 11 could be normalized, i.e., C/C0 thus it would be easier to see and analyze the photodegradation percentage.

Author Response

To Editor-in-Chief, Multidisciplinary Digital Publishing Institute (MDPI).

Processes

July 10th, 2022.        

Dear Editor,

Thank you for your favorable response to our manuscript entitled: “Enhanced adsorption, photocatalytic degradation efficiency of phenol red using CuZnAl hydrotalcite synthesized by co-precipitation technique”. Manuscript number: processes-1811153.

We would like to express our gratitude for the Editor and Reviewer’s efforts to improve the quality of this manuscript. We are herewith sending the revised manuscript which has been corrected based on the suggestions from you and the reviewers, including extensive revision in English grammar as well as a reorganized structure based on previous content.

We have highlighted in yellow to show changes and/or additions to the previous draft. We appreciate your time and effort in reviewing our manuscript and we look forward to hearing from you soon.

Best regards!

Reviewer 2 Report

In the Abstract, replace H2O2 30% to 30% H2O2

“medium pH of weak acidic range” please specify the pH value, if you mean pH 7 then say neutral pH

In the material Synthesis Cu2+ and not Cu2+ , same for N2 adsorption

“and performed as describing in the article” should be “and performed as described in the article”

Can you explain the difference between equation 3 and 4?

Is it “bicromate method” or bichromate? Please check

he results of EDX spectra of two representative material samples H, CuH-3.5 in Figure 5” its Figure 5a

“so it can exchange ions with Co32- ions” you means CO32-

Correlation between adsorption capacity and photocatalytic activity should be illustrated

Author Response

(The authors gave the same response as above.)

Reviewer 3 Report

Attached

Author Response

(The authors gave the same response as above.)

Round 2

Reviewer 1 Report

Dear authors,

Thank you for considering the reviewer's comments. Just one additional observation. In your response letter you gave a consistent response to the point 5 regarding the crystalline structure. However, such response does not appear in the revised manuscript and I consider that it would improve the quality of it. Thus, I suggest to include it.

Author Response

Dear Editor and reviewers,

We appreciate your positive review and suggestion that our manuscript may be accepted!. In the modified version. Thanks very much!

Yours sincerely,

Thi Kim Ngan Tran

Reviewer 2 Report

Comments resolved 

Author Response

(The authors gave the same response as above.)

Reviewer 3 Report

The authors have attended to all the suggested comments.

Author Response

(The authors gave the same response as above.)

Round 3

Reviewer 1 Report

Dear authors,

Thank you for considering my suggestion to include the explanation regarding the point 5.